

# Association between hand grip strength and quality of life in children with cerebral palsy: a cross-sectional study

Mshari Alghadier[1], Nada Almasoud[2], Dalia Alharthi[3], Omar Alrashdi[4] and Reem Albesher[5]

[1] Department of Health and Rehabilitation Sciences, Prince Sattam bin Abdulaziz University, Kharj, Saudi Arabia
[2] Department of Physical Therapy, Maternity and Children's Hospital in Alkharj, Kharj, Saudi Arabia
[3] Department of Physical Therapy, Alhada Armed Forces Hospital, Taif, Saudi Arabia
[4] Department of Physical Therapy, King Khalid Hospital, Hail, Saudi Arabia
[5] Department of Rehabilitation Sciences, Princess Nourah bint Abdulrahman University, Riyadh, Saudi Arabia

## ABSTRACT

**Background:** Cerebral palsy (CP) covers a wide range of causes and symptoms. It is characterized by persistent motor and postural dysfunction caused by a non-progressing pathological lesion of the immature brain. Development of fine motor skills, such as the ability to manipulate objects with smaller muscles, is crucial for a child's development. It is evident that there is a lack of hand grip strength (HGS) and quality of life (QoL) data in children with CP compared to typically developed (TD) children. Understanding the relationship between these factors might help facilitate healthcare provision and provide insight into rehabilitation programs. The aim of this study is to investigate the relationship between HGS and health-related quality of life (HRQoL) in children with CP compared to TD children.

**Methods:** An experimental cross-sectional study was conducted and 60 children (30 CP and 30 TD) were chosen; age, gender, height, weight, body mass index, preferred hand, number of siblings, school attendance, and housing type data were collected. HGS was measured using a standard hand dynamometer, and HRQoL was measured using the KIDSCREEN-10 item questionnaire.

**Results:** There was a statistically significant main effect of gender on the average HGS, $F_{(1, 56)} = 24.09$, $p < 0.001$, and the KIDSCREEN-10 sum score, $F_{(1, 56)} = 8.66$, $p < 0.001$, and the main effect of group on the KIDSCREEN-10 sum score, $F_{(1, 56)} = 17.64$, $p < 0.001$. A significant correlation between HGS and the KIDSCREEN-10 sum score in the CP group ($r = 0.35$, $p = 0.03$), and the TD group ($r = 0.56$, $p = 0.001$).

**Conclusion:** HGS was lower in children with CP, and girls had significantly lower HGS compared to boys in both groups, CP and TD children. HRQoL was significantly lower in children with CP, with boys reporting higher HRQoL on the KIDSCREEN-10 questionnaire compared to girls. Our data showed that the higher the KIDSCREEN-10 sum score is, the stronger the HGS of children in both groups. The results of this study indicate that hand grip strength may significantly impact the

Corresponding author
Mshari Alghadier,
M.alghadier@psau.edu.sa

QoL of children with CP. A correlation between HGS and HRQoL points to the importance of improving strength in children with CP through interventions and directed rehabilitation programs.

## INTRODUCTION

As an umbrella term, cerebral palsy (CP) describes a variety of etiologies and clinical manifestations. It is often described as a non-progressing pathological lesion of the immature brain that causes persistent postural and motor problems (*Bax et al., 2005*). In developed countries, CP is the most common neurological disorder associated with physical disability in children (*Michael-Asalu et al., 2019*). Many symptoms may accompany CP, including spastic paresis, ataxia, dyskinesia, impaired sensation, cognitive disorders, speech problems, visual disturbances, and epilepsy (*Bax et al., 2005*; *Odding, Roebroeck & Stam, 2006*). In recent years, several studies have indicated a decline in the prevalence of CP worldwide (*Jonsson et al., 2019*; *Robertson et al., 2017*; *Touyama et al., 2016*). Multiple studies have reported that the prevalence of CP in Saudi Arabia ranges from 0.41 to 2.3 per 1,000 live birth (*Al-Jabri et al., 2022*; *Al Salloum et al., 2011*).

The development of motor skills is crucial for children's ability to perform daily tasks, improve their physical health, and develop their cognitive skills (*Van der Fels et al., 2015*; *Bolger et al., 2021*; *Dapp, Gashaj & Roebers, 2021*). In typically developing children (TD) aged 8 to 12 years, muscle strength, coordination, and postural control usually increase as the central nervous system (CNS) matures (*Patel et al., 2020*; *Cumberworth et al., 2007*). However, children with CP commonly suffer from compromised CNS function, resulting in varying degrees of muscle weakness, spasticity, and coordination impairment (*Cappellini et al., 2020*; *Lorentzen et al., 2019*; *Paul et al., 2022*). Consequently, a lack of isometric strength can greatly affect their ability to maintain posture and perform tasks requiring stability and endurance. Due to this, children with CP may have difficulty participating in activities that facilitate motor skill development, adversely impacting their physical and social well-being, as well as their quality of life (QoL).

Children with CP have increased muscle tone due to brain injury in addition to limited hand function, coordination, and balance (*Paul et al., 2022*; *Vitrikas, Dalton & Breish, 2020*). This is thought to be due to the increased amount of muscle fibers required to accomplish a given task compared to TD children. The accumulation of collagen in myofibers has been associated with decreased muscle flexibility, and the impairment of the neuromuscular junction may affect muscle contraction (*Arnaud et al., 2021*). As a result, children with CP may experience difficulty dressing, eating, or writing, requiring assistive devices or caregiver assistance to accomplish tasks that TD children can achieve independently.

A child's fine motor skills, which include the ability to manipulate objects with their smaller muscles, are essential to their development. Fine motor skills allow children to

develop manual abilities, which later contribute to various cognitive abilities, such as writing and drawing (*Tóth, 2017*). An assessment of hand grip strength (HGS) is used to determine physical functioning, maximum voluntary force, musculoskeletal fitness, and the relationship between arm, trunk, and leg strength (*Ortega et al., 2015*; *Wong, 2016*). In both, CP and TD children, the HGS can indicate the child's overall muscle strength and physical wellness (*Savva et al., 2014*; *Peolsson, Hedlund & Öberg, 2001*). It is among the easiest and most cost-effective used measurements of physical and hand function in children with CP (*Schreuders et al., 2003*).

Health-related quality of life (HRQoL) refers to an individual's perception and subjective evaluation of their health and well-being within the context of their culture (*The WHOQOL Group, 1995*). Quality of life instruments are increasingly used as outcome measures within several settings such as clinical research, population health surveys, and clinical practice for both adults and children. A 2008 review identified 30 generic and 64 disease-specific instruments available for use with children and adolescents (*Solans et al., 2008*). Children and adolescents with CP showed similar HRQoL scores to their age-matched TD children except on measures of social participation and motor function (*Colver et al., 2015*; *Shikako-Thomas et al., 2012*; *Power et al., 2020*; *Dickinson et al., 2007*). Several factors have been identified as predictive of adverse HRQoL, including pain, parenting stress, and psychological problems (*Dickinson et al., 2007*; *Böling et al., 2013*; *Im, Cho & Kim, 2019*; *Rapp et al., 2017*). Even though motor impairment negatively affected functioning and participation, it had a much smaller impact on psychological well-being (*Mc Manus, Corcoran & Perry, 2008*).

Previous research has indicated a negative relationship between HRQoL and HGS with age in adult and older adults' population (*Halaweh, 2020*; *Musalek & Kirchengast, 2017*; *Kang, Lim & Park, 2018*). A recent study has indicated that higher HGS is associated with higher cognitive function score such as working memory, speed, attention, and self-control in adults with CP (*Heyn et al., 2023*). Performing cognitive functions is essential for learning, working, and managing daily activities that may impact HRQoL. However, little is known about the association between HRQoL and HGS for children with CP. In order to develop effective interventions and strategies to improve the general well-being of children with CP, it is essential to understand the relation between HGS and HRQoL. Therefore, the aim of this study is to investigate the relationship between HRQoL and HGS in children with CP compared to their TD counterparts. Our hypothesis was that children with CP will have lower HGS and HRQoL scores than their TD peers, along with a role for gender and age in these differences.

# MATERIALS AND METHODS

## Participants

A total of 60 age-matched children (30 CP and 30 TD), aged 8–12 years, were recruited for this experimental cross sectional study through convivence sampling (Fig. 1). There were several eligibility criteria for CP children, including the ability to walk, use their upper limbs, Gross Motor Function Classification System (GMFCS) level I to III, and a sound cognitive ability. Cerebral palsy children with severe intellectual disabilities, unable to use

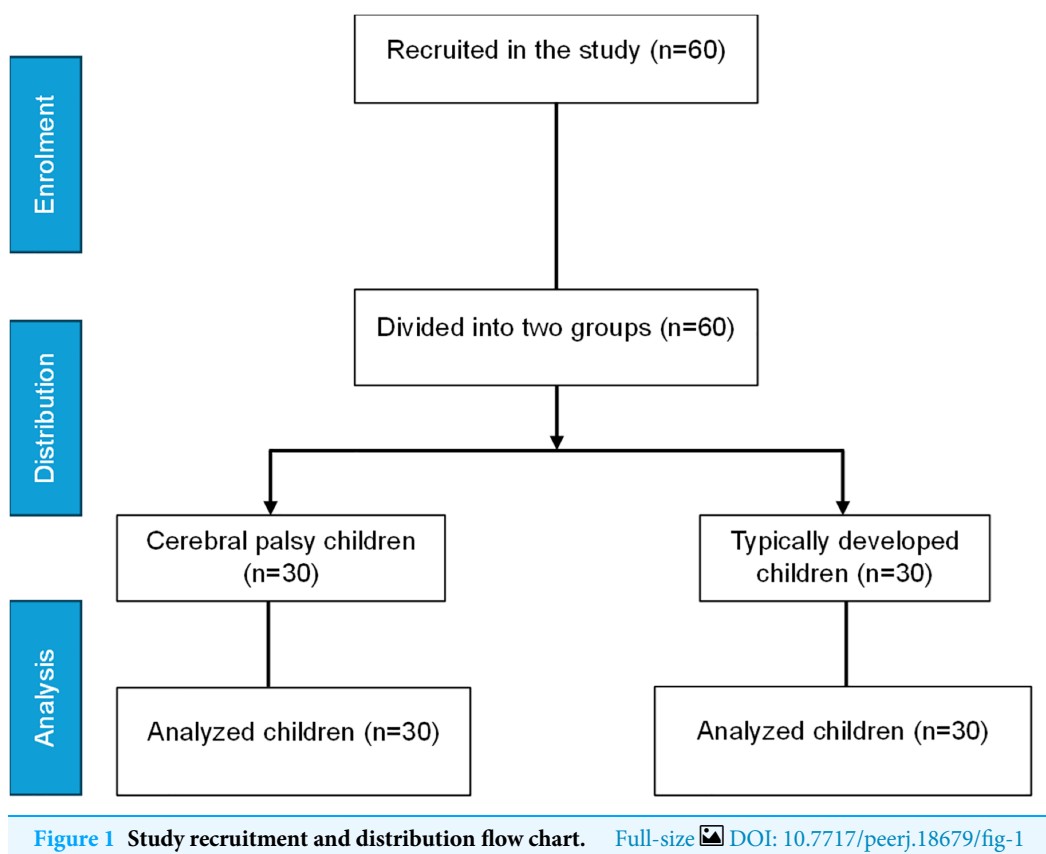

**Figure 1 Study recruitment and distribution flow chart.**

upper limbs, GMFCS level IV and level V, or who have undergone surgery within the past six months were excluded from the study. There are five levels in the GMFCS, and it's used to classify the gross motor function ability in children with cerebral palsy. In level I, the individual can walk without difficulty; however, gross motor skills are limited. In level II, the individual is able to walk without assistance; however, they are limited in their ability to walk outdoors and in public places. In level III, the individual can walk outside and in the community with the assistance of mobility devices. In level IV, children have limited mobility and are transported to and from the community and outdoors by power mobility. In level V, there are significant limitations in mobility, despite the use of assistive technology (*Rosenbaum et al., 2002*). Cerebral palsy patients were recruited from various healthcare facilities (clinics and hospitals) and TD children were recruited from different schools in Riyadh City, Saudi Arabia during the period of 1st of August 2023 to 1st of March 2024. The parents were informed of the goals, the procedures followed, and the objectives of the research study before a written informed consent was obtained.

## Procedure

During the visit, the participants had their demographic information collected; age, gender, height, weight, body mass index (BMI), preferred hand, number of siblings, school attendance, and housing type. The preferred hand was determined with an observation of activity of daily living performance such as writing, reaching, and grasping. Hand grip

dynamometer was used to measure hand muscle strength, the Arabic versions of the GMFCS Family Questionnaire utilized to determine GMFCS level, and the Arabic version of the QoL KIDSCREEN-10 questionnaire used to examine the HRQoL. All outcome measures were collected by trained pediatric physiotherapist.

### Hand grip strength

The American Society of Hand Therapists (ASHT) recommendations were followed to collect HGS by asking participants to sit on a chair facing the examiner with their feet flat on the ground. Their shoulder was neutrally rotated and adducted. The elbow was flexed at 90° with the forearm in neutral placement. The wrist was extended between 0° and 30° and ulnar deviated between 0° and 15° (*American Society of Hand Therapists, 1981*). Following verbal instructions and demonstration of the test position, three practice trials were performed alternating the preferred and non-preferred hand. Hand grip strength measured in kilograms (kg) using standard hand dynamometer (Baseline® Hydraulic Hand Dynamometer, Fabrication Enterprises, White Plains, NY, USA). The concurrent validity and inter-instrument reliability of baseline hydraulic dynamometers are satisfactory, and it's been validated and previously used to measure HGS in children with CP (*Allen & Barnett, 2011*; *Gąsior et al., 2020*; *Mandanka & Diwan, 2020*; *Dekkers et al., 2020*). Verbal encouragements (*i.e.*, squeeze as hard as you can) were introduced and children performed three trials for the preferred hand, then the mean values of the three trials were recorded. Children were given one-minute to rest between trials to minimize the effects of fatigue (*Trossman & Li, 1989*).

### KIDSCREEN-10

For children and adolescents, the KIDSCREEN QoL instrument is available in three versions: a 52-item version (*Ravens-Sieberer et al., 2005*, *2008*), a 27-item version (*Ravens-Sieberer et al., 2007*; *Robitail et al., 2007*), and an index of 10 items (*Ravens-Sieberer et al., 2010*). The KIDSCREEN-10 questionnaire is a valid and reliable, fully open access tool used to evaluate the HRQoL of children aged from 8–18 (*Ravens-Sieberer et al., 2010*). Both a self-report and a parent-report versions are available, each containing 10 items scored on a 5-point scale. Multiple studies have used the KIDSCREEN-10 questionnaire for children with CP in different cultural contexts (*Bingol et al., 2023*; *Braccialli et al., 2016*; *Maier et al., 2022*; *Milićević, 2023*). The KIDSCREEN-10 items as reported by *Ravens-Sieberer et al. (2010)* are as follow: (1) Have you felt fit and well? (2) Have you felt full of energy? (3) Have you felt sad? (4) Have you felt lonely? (5) Have you had enough time for yourself? (6) Have you been able to do the things that you want to do in your free time? (7) Have your parent(s) treated you fairly? (8) Have you had fun with your friends? (9) Have you got on well at school? (10) Have you been able to pay attention? Answer categories item 1 and 9: not at all; slightly; moderately; very; and extremely. All other items: never; seldom; quite often; very often; and always.

Items 1 and 2 explore the level of the child's/adolescent's physical activity, energy and fitness. Items 3 and 4 cover how much the child/adolescent experiences depressive moods and emotions and stressful feelings. Items 5 and 6 ask about the child's opportunities to

structure and enjoy his/her social and leisure time and participation in social activities. Item 7 explores the quality of the interaction between child/adolescent and parent or carer and the child's/adolescent's feelings toward their parents/carers. Item 8 examines the nature of the child's/adolescent's relationships with other children/adolescents. Lastly, items 9 and 10 examine a child's/adolescent's perception of his/her cognitive ability and school performance. The responses were coded so that higher values were associated with better HRQoL; the sums were then calculated, and Rasch person parameters (PP) were assigned to each possible sum score. Based on the PPs, a mean of 50 and a standard deviation (SD) of approximately 10 were calculated (*The KIDSCREEN Group Europe, 2006*). Low scores indicate poor HRQOL, whereas high scores indicate better HRQOL.

## Statistical analysis

Descriptive statistics were used to characterize the sample; frequencies and percentages were reported for categorical variables, while mean and standard deviation (SD) were reported for continuous variables. The comparisons between groups were made according to gender and group (CP and TD). An independent sample t-test and Chi-square tests were performed to determine the differences in age, gender, height, weight, BMI, number of siblings, HGS, normalized HGS, preferred hand, housing type, and school attendance between groups. Normalized HGS was calculated by dividing the absolute average HGS by body weight. Two-way analysis of variance (ANOVA) tests were performed to analyze the differences in HGS and KIDSCREEN-10 sum scores between gender and group. A Kolmogorov–Smirnov test was used to test for normality on the KIDSCREEN-10, and a Mann–Whitney U test was conducted to evaluate the difference in KIDSCREEN-10 items between groups. The correlation between the KIDSCREEN-10 sum score and HGS was assessed using Spearman's rank correlation coefficient. The strength of correlation was interpreted as recommended by *Altman (1990)*; no correlation: 0 to ±0.3, low correlation: ±0.3 to ±0.5, moderate correlation: ±0.5 to ±0.7, high correlation: ±0.7 to ±0.9, very high correlation: ±0.9 to ±1.0, and complete correlation: ±1. Statistical significance was set at $p < 0.05$, and data were analyzed using R version 4.0.3 (*R Core Team, 2020*).

## Ethical considerations

Informed consent from the participant's parents or legal guardians was acquired prior to data collection. The Declaration of Helsinki's ethical guidelines were followed when conducting this investigation. The study was approved by the Departmental Ethical Committee, Department of Health and Rehabilitation Sciences, Prince Sattam bin Abdulaziz University, Saudi Arabia (No. RHPT/023/015). All methods were performed in accordance with relevant institutional review boards and regulations. Throughout the study, participant data confidentiality and anonymity were guaranteed.

## RESULTS

A total sample of 30 CP patients and 30 TD children were included in the study. Weight, BMI, and number of siblings were statistically significantly different between CP and TD children; however, height and HGS were not statistically significant between the two

**Table 1 Demographic characteristics of the sample included divided by two groups; cerebral palsy and typically developed children.**

| Variable | Typically developed | Cerebral palsy | p-value |
|---|---|---|---|
| Age, y | 9.27 ± 1.41 | 9.53 ± 1.11 | 0.41 |
| Gender (%) | | | |
| Boys | 14 (23.3) | 14 (23.3) | 1.0 |
| Girls | 16 (26.7) | 16 (26.7) | |
| Height, cm | 127.63 ± 9.75 | 122.83 ± 11.12 | 0.08 |
| Weight, kg | 33.86 ± 10.49 | 22.78 ± 7.15 | <0.001* |
| BMI | 20.44 ± 4.41 | 14.81 ± 2.68 | <0.001* |
| Number of siblings | 3.87 ± 2.62 | 2.60 ± 1.25 | 0.02* |
| HGS 1, kg | 18.80 ± 10.78 | 15.37 ± 5.89 | 0.13 |
| HGS 2, kg | 18.53 ± 10.10 | 15.37 ± 5.94 | 0.14 |
| HGS 3, kg | 19.27 ± 10.09 | 15.67 ± 5.64 | 0.09 |
| Average HGS, kg | 18.87 ± 10.27 | 15.47 ± 5.72 | 0.11 |
| Normalized HGS, kg | 0.14 ± 0.07 | 0.12 ± 0.04 | 0.25 |
| Preferred hand n (%) | | | |
| Right | 28 (46.7) | 20 (33.3) | 0.01* |
| Left | 2 (3.3) | 10 (16.7) | |
| Housing type (%) | | | |
| House with yard | 20 (33.3) | 26 (43.3) | 0.06 |
| House without yard | 10 (16.7) | 4 (6.7) | |
| School attendance (%) | | | |
| Yes | 30 (50) | 24 (40) | 0.01* |
| No | 0 (0) | 6 (10) | |

Notes:
* $p$-value significant <0.05.
HGS, hand grip strength.
$X^2$ and independent sample t-test.

groups ($p = 0.08$, $p = 0.11$, respectively). Table 1 summarizes the demographic characteristics of the sample divided by groups.

A two-way ANOVA was conducted that compared the effect of gender and group on the average HGS and the KIDSCREEN-10 sum score. There was a statistically significant main effect of gender on the average HGS, $F (1, 56) = 24.09$, $p < 0.001$, main effect of gender on the KIDSCREEN-10 sum score, $F (1, 56) = 8.66$, $p < 0.001$, and statistically significant main effect of group on the KIDSCREEN-10 sum score, $F (1, 56) = 17.64$, $p < 0.001$. Table 2 presents the mean and standard deviation of average HGS and the KIDSCREEN-10 sum score in the sample divided by group and gender.

A *post hoc* comparison using the Tukey honestly significant difference (HSD) test revealed that the average HGS was lower in CP girls compared to TD girls, but not statistically significant ($p = 0.71$). The average HGS was lower in CP boys compared to TD boys, but not statistically significant ($p = 0.36$). A *post hoc* comparison using the Tukey HSD test revealed that the mean score of the KIDSCREEN-10 was significantly lower in CP

**Table 2 The hand grip strength and KIDSCREEN-10 sum score of the sample divided by group and gender.**

| Variable | Gender | Group | | Group × Gender |
|---|---|---|---|---|
| | | Typically developed | Cerebral palsy | |
| HGS average | Boys | 24.14 ± 9.1 | 19.76 ± 4.4 | $F = 24.09$ |
| | Girls | 14.25 ± 9.1 | 11.71 ± 3.7 | $p = 0.001^*$ |
| KIDSCREEN-10 sum score | Boys | 47.1 ± 2.18 | 44 ± 5.23 | $F = 8.66$ |
| | Girls | 45.5 ± 5.22 | 38.6 ± 4.92 | $p = 0.001^*$ |

Notes:
* $p$-value presented the significant difference between gender.
Data presented as Mean ± SD.
Two-way ANOVA.
HGS, hand grip strength

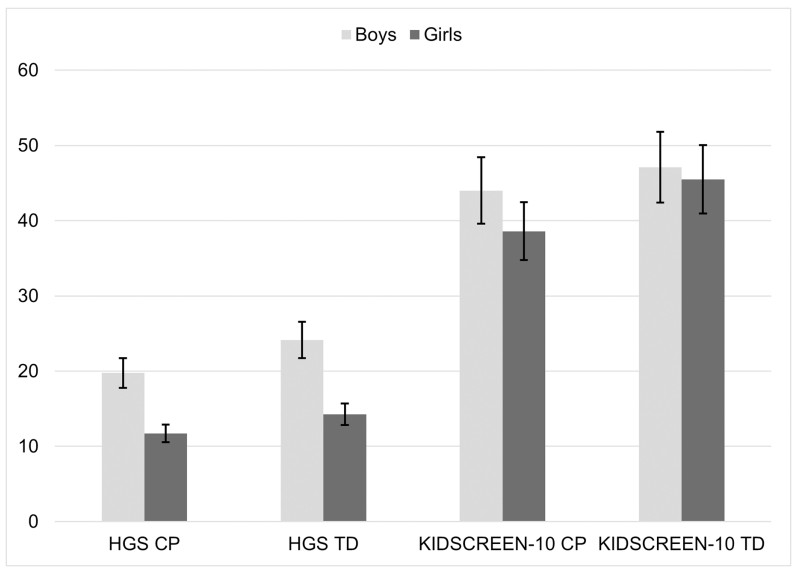

**Figure 2 Difference between groups cerebral palsy (CP) *vs* typically developed (TD) and gender (boys *vs* girls) in average hand grip strength (HGS) and KIDSCREEN-10.**

girls compared to TD girls ($p < 0.001$) and was significantly lower in CP boys compared to TD boys ($p < 0.001$) (Fig. 2).

Table 3 presents the number and percentages of the KIDSCREEN-10 items reported by parent. Overall, CP patients had lower average on the majority of HRQoL items reported by parents. They have reported a lower score on general health status question (In general, how would your child rate her/his health?) at the end of the KIDSCREEN-10 questionnaire. A Mann–Whitney U test revealed a significant difference ($U = 172$, $p < 0.001$), between the CP group (4.07 ± 0.78) and TD group (4.93 ± 0.25).

A Spearman correlation was conducted to examine the relationship between HGS and KIDSCREEN-10 sum score among both groups. A significant correlation was recorded between HGS and the KIDSCREEN-10 sum score in CP group ($r = 0.35$, $p = 0.03$), and TD

**Table 3 KIDSCREEN-10 items reported by parents divided by groups.**

| Item | Typically developed | | | | | | Cerebral palsy | | | | | |
|---|---|---|---|---|---|---|---|---|---|---|---|---|
| | Answer category | | | | | Mean ± SD | Answer category | | | | | Mean ± SD |
| | 1 | 2 | 3 | 4 | 5 | | 1 | 2 | 3 | 4 | 5 | |
| 1. Has your child felt fit and well? | 0 | 0 | 0 | 2 (3.3) | 28 (46.7) | 4.93 ± 0.25 | 4 (6.7) | 0 | 8 (13.3) | 12 (20) | 6 (10) | 3.53 ± 1.22 |
| 2. Has your child felt full of energy? | 0 | 0 | 0 | 10 (16.7) | 20 (33.3) | 4.67 ± 0.47 | 0 | 0 | 8 (13.3) | 10 (16.7) | 12 (20) | 4.13 ± 0.81 |
| 3. Has your child felt sad? | 0 | 2 (3.3) | 6 (10) | 10 (16.7) | 12 (20) | 4.07 ± 0.94 | 0 | 2 (3.3) | 12 (20) | 12 (20) | 4 (6.7) | 3.60 ± 0.81 |
| 4. Has your child felt lonely? | 0 | 0 | 6 (10) | 2 (3.3) | 22 (36.7) | 4.53 ± 0.81 | 0 | 0 | 6 (10) | 4 (6.7) | 20 (33.3) | 4.47 ± 0.81 |
| 5. Has your child had enough time for him/herself? | 0 | 0 | 0 | 12 (20) | 18 (30) | 4.60 ± 0.49 | 2 (3.3) | 2 (3.3) | 4 (6.7) | 2 (3.3) | 20 (33.3) | 4.20 ± 1.29 |
| 6. Has your child been able to do the things that he/she wants to do in his/her free time? | 0 | 0 | 2 (3.3) | 14 (23.3) | 14 (23.3) | 4.40 ± 0.62 | 2 (3.3) | 0 | 4 (6.7) | 20 (33.3) | 4 (6.7) | 3.80 ± 0.92 |
| 7. Has your child felt that his/her parent(s) treated him/her fairly? | 0 | 0 | 2 (3.3) | 2 (3.3) | 26 (43.3) | 4.80 ± 0.55 | 0 | 0 | 0 | 10 (16.7) | 20 (33.3) | 4.67 ± 0.47 |
| 8. Has your child had fun with his/her friends? | 0 | 0 | 0 | 4 (6.7) | 26 (43.3) | 4.87 ± 0.34 | 0 | 2 (3.3) | 0 | 12 (20) | 16 (26.7) | 4.40 ± 0.81 |
| 9. Has your child got on well at school? | 0 | 0 | 2 (3.3) | 6 (10) | 22 (36.7) | 4.67 ± 0.60 | 0 | 6 (10) | 4 (6.7) | 4 (6.7) | 16 (26.7) | 4 ± 1.23 |
| 10. Has your child been able to pay attention? | 0 | 0 | 2 (3.3) | 4 (6.7) | 24 (40) | 4.73 ± 0.58 | 0 | 0 | 6 (10) | 8 (13.3) | 16 (26.7) | 4.33 ± 0.80 |
| In general, how would your child rate her/his health? | 0 | 0 | 0 | 2 (3.3) | 28 (46.7) | 4.93 ± 0.25 | 0 | 0 | 8 (13.3) | 12 (20) | 10 (16.7) | 4.07 ± 0.78 |

**Notes:**
Data presented as number (percentage) and mean ± SD.
Answer categories item 1 and 9: (1) not at all; (2) slightly; (3) moderately; (4) very; and (5) extremely. All other items: (1) never; (2) seldom; (3) quite often; (4) very often; and (5) always.

**Table 4 Correlation between hand grip strength and KIDSCREEN-10, age, and number of siblings divided by groups.**

| Variables | HGS typically developed | | HGS cerebral palsy | |
|---|---|---|---|---|
| | r Spearman | $p$-value | r Spearman | $p$-value |
| KIDSCREEN-10 sum score | 0.56 | <0.001** | 0.35 | 0.03* |
| Age | 0.49 | <0.001** | 0.15 | 0.40 |
| Number of siblings | 0.33 | 0.07 | 0.14 | 0.45 |

**Notes:**
* Correlation is significant at the <0.05 level.
** Correlation is significant at the 0.01 level.

group ($r = 0.56$, $p = 0.001$). Table 4 summarizes the correlation coefficient between HGS and KIDSCREEN-10 sum score, age, and number of siblings.

## DISCUSSION

In this experimental cross-sectional study, we have investigated the relationship between HGS and HRQoL in children with CP compared to TD counterparts. Our results showed

that HGS was lower in CP children, and girls had significantly lower HGS compared to boys in both groups, CP and TD children. Moreover, normalized HGS was greater in TD children compared to CP peers. Health-related quality of life was significantly lower in children with CP, with boys reporting higher HRQoL on the KIDSCREEN-10 questionnaire compared to girls. Our data showed that the higher KIDSCREEN-10 sum score is, the stronger HGS of children in both groups.

It is not a surprise that HGS in children with CP is lower than their TD counterparts due to the neurological impairments accompanying the brain damage in CP patients. Children with CP have a lower normalized HGS than TD children, suggesting that HGS is influenced by body mass, rather than muscle morphology or neural function. Children with higher body mass may have a mechanical advantage, allowing for stronger grip even if their muscle quality or neural function is not superior. Thus, differences in body mass can lead to variations in HGS between individuals (*Rostamzadeh et al., 2021*; *Alkholy, El-Wahab & Elshennawy, 2017*). Consequently, muscle strength evaluations should not rely solely on HGS measurements but also consider the child's body weight to avoid misinterpretations. Additionally, alternative methods that assess muscle quality and neural function should be employed to gain a comprehensive understanding of a child's physical development.

It has long been known that hand strength has an important role to play in activity participation and functional ability of children with CP (*Brown et al., 1987*). Musculoskeletal impairments, sarcopenia, and chronic pain are common in CP patients that make it difficult for them to perform activities of daily living (*Stavsky et al., 2017*; *Heyn et al., 2019*; *Peterson, Gordon & Hurvitz, 2013*). Research into the relationship between hand impairments and manual ability has indicated that motor functions such as grip strength, dexterity, and spasticity seem to be related to manual ability (*Cans, 2000*; *Verschuren et al., 2007*). In comparison with previously published normative data for TD children, our HGS data are slightly higher. Data collection procedures, sample size, and devices used may have contributed to this result (*Alqahtani et al., 2023*; *Omar, Alghadir & Al Baker, 2015*).

A variety of physiological factors, including muscle mass, hormonal differences, and varying levels of physical activity, contribute to gender differences in HGS, with boys typically showing greater strength than girls (*Rostamzadeh et al., 2021*; *Górecki et al., 2024*). Studies have found that boys experience a greater increase in muscle strength during puberty than girls, as these differences emerge early and become more pronounced during childhood (*Al-Rahamneh et al., 2020*; *Wen et al., 2020*). In children with CP, spasticity and muscle atrophy may differ by gender and may impact HGS. In line with our findings, previous studies have indicated that boys with CP may experience greater muscular development despite the condition, potentially resulting in greater grip strength than girls with CP, who might experience greater muscle weakness and coordination difficulties (*Mehmood et al., 2023*; *Dekkers et al., 2020*).

Although it is difficult to quantify QoL for a chronic and multifaceted condition such as CP, several studies have investigated this relationship (*Dickinson et al., 2007*; *Bjornson et al., 2008*; *Du et al., 2010*; *Calley et al., 2012*; *Vles et al., 2015*). In a recent systematic

review published in 2019, eleven studies were included to compare the QoL of children and adolescents with CP compared to their TD peers. Overall, CP patients experienced limited ability to participate in physical activities such as self-care, exercise, and poor physical health (*Makris, Dorstyn & Crettenden, 2021*). There has also been a report of lowered self-esteem, poor mental health, and a negative impact on the parents' own wellbeing as a result of their child's disability. In two studies using the KIDSCREEN tool, parents have reported a higher QoL in relation to parent-child relationship and life at home. Further, they have reported a positive perception of the school environment, including engagement and learning (*Dickinson et al., 2007*; *Wake, Salmon & Reddihough, 2003*). As compared to our findings, we found similar results regarding limited QoL for children with CP compared to those with TD. Although these studies included a larger sample size, they also examined the differences between parent-reported and child-reported questionnaires. Parent-reported questionnaires are especially important when treating children with severe CP symptoms due to intellectual and/or communicative disabilities. It should be noted, however, that parent-reported questionnaires may introduce systematic bias as they are based on indirect cues and personal experience (*Cummins, 2002*).

Research indicates that gender can have an impact on the perception and experience of HRQoL in TD children (*Chen et al., 2020*). Different socialization patterns, expectations, and coping strategies among boys and girls can influence their mental and physical health. According to our study, boys with CP have higher HGS, suggesting that they have greater muscle strength and physical activity as compared to girls, which may be beneficial to their physical well-being. On the other hand, girls with CP may experience greater limitations in mobility and strength, which can lead to lower scores in HRQoL physical domains (*Makris, Dorstyn & Crettenden, 2021*; *Sharawat & Panda, 2022*). In accordance with our findings, recent studies have suggested that boys with CP may experience advantages in physical functioning and social support, leading to higher QoL scores compared to girls with CP (*Park et al., 2016*; *Radsel, Osredkar & Neubauer, 2017*; *Tedla et al., 2024*).

To our knowledge this is the first study to evaluate the relationship between the HGS and HRQoL in children with CP and their TD counterparts. Hand function is crucial to carry out day-to-day activities such as eating and dressing. Children with CP face these challenges frequently, which may negatively influence their activity level and social participation. We have presented a positive correlation between the KIDSCREEN-10 sum score and HGS in both children with CP and TD children. This correlation indicated that HGS influences different aspects of HRQoL, such as physical activity, depressive moods and emotions, social participation and leisure time, the quality of interaction between the child and their parents, and finally the cognitive ability perception. These findings serve as a baseline of future research which could further shed light into the influence of HGS on HRQoL of children with CP with different phenotypes, gross motor level, and manual ability level.

There are several limitations to this study. First, the cross-sectional study design does not prove a causal connection between HGS and HRQoL. A prospective study is necessary to examine the relationship between low HGS and associated variables in the future. Second, the limited sample size compared to previously published data suggests that

caution should be exercised when interpreting these findings. Third, although the HGS tool used in the current study is easy to administer, cost-effective, and practical, its psychometric properties might concern some researchers. Future research should therefore use a more valid and reliable tool. Fourth, the KIDSCREEN-10 item is valid, reliable, and takes a few minutes to complete. However, its limited items, in contrast to the larger versions of the KIDSCREEN-52 and KIDSCREEN-27, prevented us from covering all HRQoL dimensions in this study. Finally, the generalizability of our conclusions to the wider CP population, such as those with GMFCS levels IV and V, is limited, hence caution should be exercised when interpreting the results.

## CONCLUSION

Overall, the results of this study indicate that HGS may significantly impact the HRQoL of children with CP. It was found that children with CP exhibited significantly lower levels of HGS than their TD peers. Disparities in this regard may have an adverse impact on their daily lives, negatively affecting their independence and participation in activities. A correlation between HGS and HRQoL points to the importance of improving strength in children with CP through interventions and directed rehabilitation programs. In order to improve the QoL for these children, it may be necessary to implement targeted interventions, such as occupational therapy or specialized exercises. Researchers must continue to explore the long-term impacts of interventions on QoL and HGS in both populations, ensuring that strategies are evidence-based and tailored to meet the individual needs of children with CP.

### Funding
This work is funded by the King Salman Center for Disability Research through Research Group No. KSRG-2023-475. The funders had no role in study design, data collection and analysis, decision to publish, or preparation of the manuscript.

### Grant Disclosures
The following grant information was disclosed by the authors:
King Salman Center for Disability Research: KSRG-2023-475.

### Competing Interests
The authors declare that they have no competing interests.

### Author Contributions
- Mshari Alghadier conceived and designed the experiments, performed the experiments, analyzed the data, authored or reviewed drafts of the article, and approved the final draft.
- Nada Almasoud conceived and designed the experiments, prepared figures and/or tables, authored or reviewed drafts of the article, and approved the final draft.
- Dalia Alharthi performed the experiments, analyzed the data, prepared figures and/or tables, authored or reviewed drafts of the article, and approved the final draft.

- Omar Alrashdi performed the experiments, analyzed the data, prepared figures and/or tables, and approved the final draft.
- Reem Albesher conceived and designed the experiments, authored or reviewed drafts of the article, and approved the final draft.

## Human Ethics

The following information was supplied relating to ethical approvals (*i.e.*, approving body and any reference numbers):

The study was approved by the Departmental Ethical Committee, Department of Health and Rehabilitation Sciences, Prince Sattam bin Abdulaziz University, Saudi Arabia (No. RHPT/023/015).

## Data Availability

The raw data is available at figshare: Alghadier, Mshari (2024). HGS and QoL in CP and TD children: figshare. Dataset. https://doi.org/10.6084/m9.figshare.27241548.v1.

## Supplemental Information

Supplemental information for this article can be found online at http://dx.doi.org/10.7717/peerj.18679#supplemental-information.

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
