# Peer review of "Association between hand grip strength and quality of life in children with cerebral palsy: a cross-sectional study"

_PeerJ, doi:10.7717/peerj.18679_

## Round 0.1 · original submission · Major Revisions

The reviewers have provided some good suggestions that I suggest you look at taking on board that will improve the quality of this manuscript. I would hope you can use their suggestions to resubmit a high-quality resubmission within the required timeline.

Reviewer 1 ·

Basic reporting

The article meets the ethical standards regarding the included subjects and applied tests in accordance with the Declaration of Helsinki

However, The article does not comply with the criteria of the journal. There are a lot of methodological flaws.

Experimental design

1. The introduction needs to be written in a little more detail and expanded with 2-3 paragraphs from the aspect of motor skills (primarily static strength), in children aged 8-12 years and CNS functioning in children with CP and typically developed children. Look for more research on this topic.

2. In the part of the Participants, the average values of height, weight, BMI of all respondents by age and gender should be defined.

3. Of the representative sample of both groups (115,116 respondents), only about 30% participated in the research (30+30). Give reasons.


4. In the Materials and Methods chapter, it would have been much clearer if the STORB diagram had been presented. This is a bit confusing.


5. In the statistical procedures, how did you determine the normality of the distribution of the results (Ks or SW test) on the basis of which you chose the multivariate Mann-Whitney U analysis?

6. The first paragraph in the results, which defines the average results of the respondents, should be listed in the Participants chapter. I think it is superfluous to establish differences in anthropological status among children.

7. In Table 1, it is not necessary to determine differences in anthropological status (Height, Weight, BMI). Enter these values in the Participants section

8. Tables do not contain results appearing (presented) in the text to which the authors refer (Two-way anova, Tukey HSD, F value?, etc.). Many badly formatted tables and very confusing for the reader.

9. The research referred to by the authors in the text is much older, 20-25 years (about 40% of the references).

Validity of the findings

Considering the implemented methodology, I am not able to assess the validity of the results.

Reviewer 2 ·

Basic reporting

Ovreall mostly clear and unambiguous.
Minor grammatical errors on lines 95 and 95
"Previous research has indicated a negative...
Recent studies have indicated...

Experimental design

Clarify if TD children were selected as age-matched controls to TD children.
Spell out GMFS levels and define- justify relevance to sampling

Procedure- for each outcome measure, include citation, justification and published psychometric properties for each (relevant to children with CP and TD).

Validity of the findings

Findings related to gender differences should be commented on in the discussion in more detail.
Findngs related to between group differences between CP group and TD group overall on HGC and Kidscreen QOL measures should be presented more cleary and concisely as this is the most relevant result in relation to the reseach question.

---

## Round 0.2 · Minor Revisions

I think the authors for their dedication in how they have taken on board the reviewers feedback in the resubmission. I’m happy to recommend acceptance of this manuscript pending the final minor edits requested below, with the line numbers reflecting the track changes version.

Line 23: “is crucial for a child development” should be written as “is crucial for a child’s development”.

Line 68: “static strength” may be better written as “isometric strength” here and anywhere else in the manuscript as this is the much more commonly used term.

Line 224 – 231: I think you can remove the actual values for things such as age, body mass and BMI in this section as they are already included in table 1.
Overall results and discussion: while the TD has substantially and significantly greater handgrip strength than children with cerebral palsy, the TD children are also substantially and significantly heavier. As handgrip strength is positively correlated to bodyweight and muscle mass, I therefore suggest you also calculate normalised hand grip strength (i.e. handgrip strength divided bodyweight) for each individual and then report such data for the different groups in Table 1. Based on the pronounced differences in body mass between these two groups, it would appear to be that the difference in handgrip strength between the two groups is predominantly a reflection of differences in body mass, rather than intrinsically different muscle morphology or neural function characteristics. I suggest such additions would need to be included in the current lines 279 – 306.

Reviewer 2 ·

Basic reporting

All feedback provided by reviewers has been addressed in this revision.

Experimental design

All feedback provided by reviewers has been addressed in this revision.

Validity of the findings

All feedback provided by reviewers has been addressed in this revision.

Additional comments

All feedback provided by reviewers has been addressed in this revision.

---

## Round 0.3 · accepted · Accept

Thanks for addressing these comments